# Use of Urine N-Terminal Prohormone of Brain-Natriuretic Peptide (NT-proBNP) as a Non-Invasive Indicator for Renal Function Recovery after Surgical Relief of Hydronephrosis

**DOI:** 10.3390/diagnostics13020247

**Published:** 2023-01-09

**Authors:** Chia Min Liu, Chan Jung Liu, Ze Hong Lu, Ho Shiang Huang

**Affiliations:** 1National Cheng Kung University Hospital, Tainan 704, Taiwan; 2Department of Urology, College of Medicine, National Cheng Kung University, Tainan 704, Taiwan

**Keywords:** cardiorenal syndrome, NT-proBNP, obstructive uropathy, biomarkers, renal function, ureterorenoscopic surgery

## Abstract

Cardiorenal syndrome is rarely discussed in patients with obstructive uropathy. On the other hand, there is currently no accurate and convenient clinical biomarker to predict the recovery of renal function after the resolution of ureteral obstruction. The purpose of this study is to explore the association between hydronephrosis and cardiorenal syndrome by measuring the change of the N-terminal prohormone of brain-natriuretic peptide (NT-proBNP), which is a biomarker typically used for cardiac failure, in patients receiving surgery to relieve obstructive uropathy. A total of 212 patients admitted for ureteroscopic (URS) procedures to relieve hydronephrosis were enrolled in this study. The severity of hydronephrosis as well as plasma and urine NT-proBNP levels were obtained before and after surgery. The results showed a significant correlation between urine NT-proBNP levels and renal function recovery following the resolution of hydronephrosis (OR 3.24, 95% CI 1.09–9.70, *p* = 0.035). Urine NT-proBNP could even predict the recovery of renal function with an area under the ROC = 0.775 (0.65–0.88, *p* < 0.001). In conclusion, urine NT-proBNP could be a useful early marker of renal function recovery after URS surgery, identifying patients whose renal and heart functions were compromised by the obstruction.

## 1. Introduction

Obstructive uropathy is one of the most common urological diseases and is frequently characterized by hydronephrosis. The most common causes of acute hydronephrosis are ureteral stones and ureteral strictures [1], which can be managed by ureterorenoscopic (URS) surgery. Theoretically, renal function should improve after the resolution of obstructive uropathy. However, recovery might not be observed in all patients for various reasons, including the chronic nature of obstruction or kidney injury caused by reasons other than post-renal factors. There is currently no accurate and convenient clinical biomarker to predict the recovery of renal function after the resolution of ureteral obstruction.

The cardiovascular and renal systems operate closely to maintain blood volume and hemodynamic stability. In recent years, this bidirectional relationship between the heart and the kidney has been widely recognized as “cardiorenal syndrome’’ (CRS), which comprises several conditions involving the failure of one organ with the consequent escalation of pathological changes in the other [2]. It is indubitable that CRS could occur due to the acute hydronephrosis-induced renal injury. Our previous study found that relieving the hydronephrosis in patients with urolithiasis resulted in improved left ventricular relaxation, enhanced left atrial function, reduced plasma N-terminal prohormone of brain-natriuretic peptide (NT-proBNP) levels, and increased NT-proBNP urinary excretion. However, few studies have addressed the importance of CRS in patients with acute hydronephrosis, and further research on CRS caused by acute hydronephrosis is lacking [3].

We conducted a prospective study to enroll patients with unilateral hydronephrosis to evaluate the changes in renal function and levels of NT-proBNP in plasma and urine after resolution of hydronephrosis by URS surgery. Due to the diagnostic role of NT-proBNP in assessing heart function [4], we measured the changes in the NT-proBNP levels to explore the association between hydronephrosis and CRS. Finally, this study also investigated the merits of using urine NT-proBNP as a non-invasive and easily accessible marker to predict the recovery of kidney function after the resolution of hydronephrosis.

## 2. Materials and Methods

### 2.1. Study Participants and Data Collection

The data for this prospective study were collected from a single tertiary medical center between September 2012 and June 2013. The protocol of this prospective study was approved by the Institutional Ethics Review Board of the National Taiwan University Hospital (Registry Number: 201205117RIC), and all participants provided written, informed consent. The enrolled participants were admitted for surgical intervention to relieve unilateral obstructive uropathy via ureteroscopic lithotripsy (URSL) or ureteroscopic balloon dilatation. We excluded patients with a history of cardiovascular disease (including heart failure), preoperative implantation of an indwelling nephrostomy tube, known genitourinary tract malignancy, acute infection, other inflammatory diseases, or patients younger than 18 years of age. 

Before patient enrollment, a sample size calculation was done. The sample size in each group was calculated to be 64, respectively, under the power of 0.8. Baseline clinical characteristics as well as plasma and urine (one-spot urine sampling) specimens were collected from all enrolled patients on the day of admission (shown in Figure 1). An experienced urologist performed preoperative renal sonography on the day of admission to determine the grade of hydronephrosis as per the following definition: grade I—slight blunting of calyceal fornices; grade II—obvious blunting of calyceal fornices and the enlargement of calices, but intruding shadows of papillae easily seen; grade III—rounding of calices with obliteration of papillae; and grade IV—extreme calyceal ballooning [5]. In the final analysis, we categorized all participants into low (grades I and II) (*n* = 108) and high (grades III and IV) grades (*n* = 103) of hydronephrosis.

After surgery, blood and urine samples were collected on postoperative day 1 (defined as ‘’immediate’’) and at 2–4 weeks after the surgery (defined as ‘’long-term’’). All participants also underwent a renal sonographic examination when they returned to the clinic 2–4 weeks after the surgery to determine the grade of hydronephrosis without ureteral stents in situ. Urine NT-proBNP(pg/mL) was corrected for urine creatinine (mg/dL) before statistical analysis and was presented as “urine NT-proBNP” (pg/mg Cr) in the Results and Discussion Sections.

### 2.2. Measurement of Biomedical Variables

For plasma samples, whole blood was taken on each occasion. The serum samples were separated and stored at −85 °C. The analysis of serum samples took place in batches within 3–6 months of collection. The urine samples were refrigerated at −20 °C immediately after collection and were analyzed after the collection of the whole cohort.

Serum BUN and creatinine were measured, and eGFR was subsequently calculated with the MDRD formula: 186 × (Serum Cr)^−1.154^ × (age)^−0.203^ × (0.742 if female) × (1.210 if African-American). NT-proBNP levels were analyzed in duplicate with a chemiluminescent immunoassay kit using an Elecsys^®^ 2010 analyzer (Roche Diagnostics, Basel, Switzerland). The intra- and interassay coefficients of variation were 10%.

### 2.3. The Procedure of the URS

An URS with either lithotripsy or balloon dilatation under general anesthesia was performed by an experienced urologist. A 6F/7.5F semi-rigid URS (Richard Wolf Medical Instruments Corporation, Vernon Hills, IL, USA) was introduced retrogradely into the ureter for lithotripsy or balloon dilatation. Normal saline irrigation through the URS was performed using a hand-held syringe to enhance instrument passage and maintain clear vision. Each patient underwent a 7 Fr. ureteral stent placement (Cook Medical, Bloomington, USA) post-operatively, which was left in place for two weeks.

### 2.4. Statistical Analysis

Continuous variables are presented as mean ± standard deviation (SD), whereas categorical variables are presented as frequencies. A Kolmogorov–Smirnov test was performed to test the normality of the distribution of the variables. A paired *t*-test was used to compare baseline and postoperative measurements. We divided the patients into those with decreased or unchanged eGFR and those with increased eGFR. Variables were compared between the two groups with either a t-test or a Mann–Whitney U test, depending on the normality of the distribution. Uni- and multi-variate logistic regressions were performed to identify the factors affecting the recovery of renal function after surgery. Variables considered confounding factors for NT-proBNP concentration were corrected with binary logistic regression [6]. Urine NT-proBNP and plasma NT-proBNP were categorized into low and high with a cutoff of 10 pg/mg Cr and 0.2 ng/mL for binary logistic regression. A Pearson correlation coefficient was used to examine the relationship between changes in renal function and variables. Finally, a receiver operator curve (ROC) was constructed to determine the predictive values of urine NT-proBNP for the recovery of renal function, with the area under the curve (AUC) calculated. Statistical analyses were performed using SPSS (SPSS v23.0 https://spss.datasolution.kr/main/main.asp (accessed date on 16 November 2022)). In all tests, a *p*-value < 0.05 was considered statistically significant.

## 3. Results

Of the 212 subjects recruited for final analysis, the average age was 54.5 years, and 63.7% were male (Table 1). Nearly half of the patients (48.5%) had grade 2–3 hydronephrosis before surgery. The average preoperative eGFR was 82.3 mL/min/1.73 m^2^ and 21.7% of patients initially presented with impaired renal function (eGFR < 60 mL/min/1.73 m^2^). 

The baseline characteristics were compared between the low- and high-grade hydronephrosis groups. We found that patients with a higher grade of hydronephrosis presented with poorer renal function and significantly lower urine NT-proBNP levels.

We further evaluated the correlation between preoperative eGFR, plasma NT-proBNP level, and urine NT-proBNP. No significant association between preoperative eGFR and either plasma or urine NT-proBNP levels was found. Plasma NT-proBNP levels were inversely correlated with urine NT-proBNP levels, as shown in Figure 2.

The values of eGFR, urine output, urine NT-proBNP and plasma NT-proBNP before and after surgery are presented in Table 2.

Although eGFR improved significantly two weeks after surgery (*p* = 0.004), no immediate difference was noted after URS. While no obvious change in plasma NT-proBNP was observed, urine NT-proBNP significantly increased (*p* < 0.001) after surgery. Multiple variables were compared between the patients with improved and deteriorated renal function (Table 3). Patients with improved renal function after URS had a significantly lower baseline eGFR (*p* < 0.001). There was a non-significant trend of decreasing plasma NT-proBNP levels in patients with improved renal function postoperatively and an increasing trend of plasma NT-proBNP levels in patients with deteriorated renal function.

In patients with long-term renal function improvement after surgery, their urine NT-proBNP at baseline was remarkably lower (*p* = 0.006) than that of those with deteriorated renal function, shown in Figure 3. Moreover, the preoperative urine NT-proBNP level was negatively correlated with the change in eGFR after hydronephrosis is relieved (*p* = 0.014). Figure 2b demonstrated the significant association between lower urine NT-proBNP and improved renal function after surgery.

Table 4 shows the results of logistic regression analyses to determine the factors associated with immediate renal function recovery after URS surgery. Univariate regression analysis revealed that preoperative CKD and urine NT-proBNP levels were associated with recovery of renal function. After adjustment, only the baseline urine NT-proBNP level was significantly associated with renal function recovery (OR 3.24, 95% CI 1.09–9.70, *p* = 0.035). Furthermore, ROC analysis was performed to assess the ability of preoperative urine NT-proBNP to predict the improvement of renal function after URS surgery, which showed an area under the ROC curve (AUC) of 0.775 (0.65−0.89, *p* < 0.001), shown in Figure 4.

## 4. Discussion

In this prospective study, we observed that a lower preoperative urinary NT-proBNP level was significantly associated with the recovery of renal function following URS, aimed at relieving hydronephrosis, even after correction for hydronephrosis and CKD. Moreover, the level of preoperative urine NT-proBNP could be a fast and accurate indicator for the recovery of renal function after relieving obstructive uropathy.

### 4.1. Plasma NT-proBNP, Urine NT-proBNP, and Hydronephrosis

NT-proBNP is one of the cardiac natriuretic peptides that is secreted by the left ventricle in response to ventricular stress, intracardiac pressure, volume overload, and ischemic injury to facilitate cardiac homeostasis via effects on vascular tone and natriuresis [4]. Plasma NT-proBNP levels increase with the severity of heart failure; hence, its plasma concentration is currently regarded as a diagnostic and prognostic tool for cardiovascular diseases and was incorporated into major guidelines for recognizing and managing heart failure. [7] As for urine NT-proBNP, its clinical utility as a non-invasive tool to reflect heart failure and cardiovascular function has also been suggested in recent years, and it was frequently corrected with a urine creatinine level before further interpretation [8,9,10]. In the present study, most patients with hydronephrosis had plasma NT-proBNP levels higher than the reference level for individuals younger than 75 years of age without heart failure, which is 125 pg/mL (0.125 ng/mL) [11]. To interpret this significant finding, it is essential to determine whether renal function compromises the accuracy of NT-proBNP levels. Although a few studies have mentioned that plasma NT-proBNP levels would be higher in patients with poor renal function [12], Tsutamoto (2019) found that plasma NT-proBNP concentration is unaffected by eGFR in non-CKD patients with an eGFR > 60 mL/min/1.73 m^2^ [13]. Meanwhile, urine NT-proBNP levels have been considered unrelated to eGFR in most studies [14,15]. To summarize, renal function may not affect the results of plasma or urine NT-proBNP levels unless the renal function is severely impaired. In our study, no significant association between renal function and plasma NT-proBNP (*p* = 0.0603) or urine NT-proBNP (*p* = 0.0596) levels was found by a Pearson correlation coefficient test. Moreover, CKD patients comprise only a small portion of our present cohort [16]. 

Regarding our results that urine NT-proBNP levels could predict renal function recovery, a bias should be considered. It is possible that the renal function of enrolled patients presented a normal eGFR level at baseline and did not show significant improvement, but others presented a lower eGFR level and had increased eGFR after surgery. In order to strengthen our findings, we have performed a logistic regression to prove that the association between urine NT-proBNP level and postoperative renal function recovery remained significant after adjusting for the presence of CKD. Taken together, we believe our findings could possibly be related to the cardiorenal syndrome caused by obstructive uropathy.

### 4.2. Cardio-Renal Syndrome and Obstructive Uropathy

CRS refers to a condition wherein the functional impairment of the heart or the kidney leads to the dysfunction of the other due to the complex bidirectional relationship between them. Specifically, type 3 cardiorenal syndrome refers to cardiac dysfunction resulting from an acute kidney injury. The possible mechanisms include the release of inflammatory mediators, oxidative stress, cellular apoptosis under renal ischemia and injury, hemodynamic imbalance, and the RAAS system [17].

However, few studies have investigated CRS following obstructive nephropathy. In previous studies exploring the relationship between obstructive nephropathy and the cardiovascular system, hydronephrosis was found to not only cause hypertension [18] but also play a role in the development of cardiomyopathy [19,20,21].

To the best of the authors’ knowledge, our previous study was the first one to evaluate cardiac function before and after the relief of obstructive nephropathy. We found that obstructive nephropathy caused diastolic dysfunction, and the resolution of hydronephrosis resulted in improved left ventricular relaxation, reduced plasma NT-proBNP levels, and increased urinary excretion of NT-proBNP [3]. The exact mechanism of CRS after obstructive nephropathy is unclear, but theoretically, it is possible that the improved microcirculation and decreased stress on the heart could ameliorate cardiac function as the hydronephrosis resolves. In this study, we also found a decreasing trend in plasma NT-proBNP levels and a significant elevation of urine NT-proBNP levels in patients whose renal function improved after the surgery to relieve the hydronephrosis. The changes in NT-proBNP levels could be explained by the recovery of heart function along with the resolution of hydronephrosis.

### 4.3. The Recovery of Renal Function after Relieving Hydronephrosis

Theoretically, renal function should improve after the resolution of obstructive uropathy. However, recovery might not be observed in all patients for various reasons, including the chronic nature of obstruction or kidney injury acting in conjunction with reasons other than post-renal factors [22]. Whenever a patient with hydronephrosis presented, it was difficult to determine whether the patient’s renal function would benefit from URS surgery in the first place. Currently, there are only a few studies on how renal function reacts to the URS procedure [23]. Methods for identifying patients with better recoverability after the relief of obstruction have rarely been discussed [24]. We believe that if the renal function has acutely and truly deteriorated due to hydronephrosis, these subjects should have improved renal function after relieving the obstruction. As obstruction of the ureteral tract develops, the renal tubule is injured, and the microcirculation is compromised. In addition, cardiac function could be influenced by the aforementioned cardiorenal syndrome, and renal plasma flow might decrease due to afferent and efferent arteriolar vasoconstriction [25]. In our study, patients whose renal function improved after URS exhibited poorer preoperative renal function, significantly lower urine NT-proBNP concentrations, and a decreasing trend in plasma NT-proBNP levels, post-operatively. Moreover, a lower urine NT-proBNP level is not only associated with but even predictive of the recovery of renal function after CKD correction. Although the exact mechanism behind this finding warrants further exploration, we hypothesize that the lower urine NT-proBNP levels in these patients are probably associated with compromised renal plasma flow due to an obstruction, causing a decreased excretion of NT-proBNP into the urine. We believe that we could thereby identify patients whose renal function is truly compromised by the obstruction and whose renal function could benefit from the URS procedure aimed at relieving hydronephrosis.

### 4.4. Limitations

This study had several limitations. First, although all participants visited our clinics with acute symptoms and were proven to have hydronephrosis on sonography, the exact duration of the hydronephrosis was unknown, which may greatly affect the results and be a significant source of bias. Nevertheless, this condition is close to the clinical scenario in which the accurate period of hydronephrosis is unknown when we encounter a patient. Second, multiple factors could interfere with the plasma and urinary NT-proBNP levels, including DM, anemia, etc. [6], which were not recorded in our cohort. Third, the evaluation of the lab data and sonography at 2–4 weeks may not be representative of the long-term condition of the patient. However, we believe that the renal function and hydronephrosis should be at a stable status at this point, which reflects the timespan when patients return to the clinic after surgery in the real world.

## 5. Conclusions

In conclusion, we found that the severity of hydronephrosis was associated with poorer renal function, lower urine NT-proBNP levels, and higher plasma NT-proBNP levels preoperatively. Moreover, the urinary NT-proBNP level may be useful as a non-invasive and early marker of renal function recovery after URS surgery, identifying those whose renal and heart functions were compromised by the obstruction.

## Figures and Tables

**Figure 1 diagnostics-13-00247-f001:**
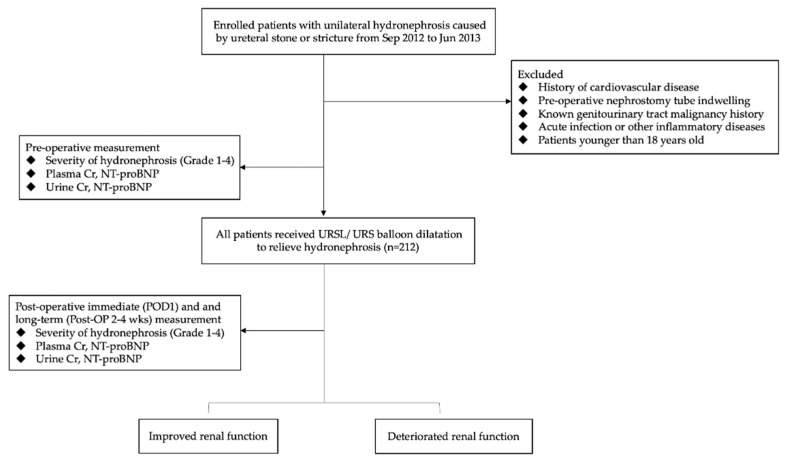
Consort flow diagram.

**Figure 2 diagnostics-13-00247-f002:**
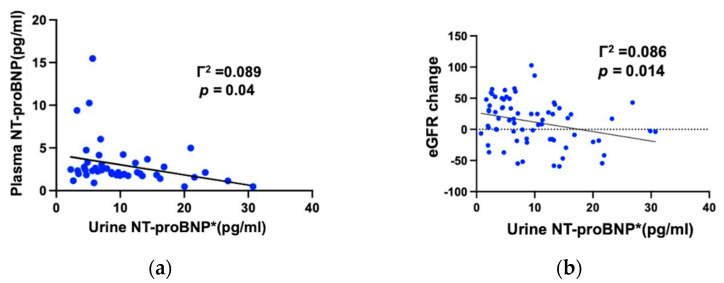
(**a**) Plasma NT-proBNP levels were inversely correlated with urine NT-proBNP levels. (**b**) The change in eGFR is inversely correlated with urine NT-proBNP levels. The lower urine NT-proBNP levels were significantly associated with improved renal function after the surgery.

**Figure 3 diagnostics-13-00247-f003:**
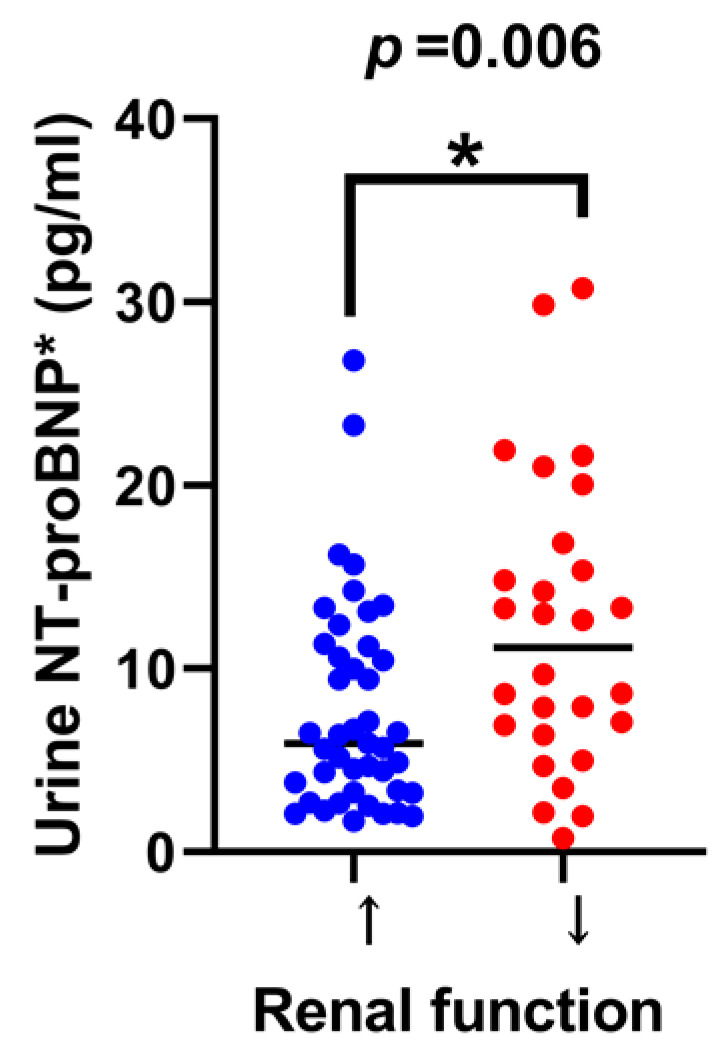
The comparison of urine NT-proBNP levels between patients with an improved renal function and a deteriorated renal function. * = statistically significant.

**Figure 4 diagnostics-13-00247-f004:**
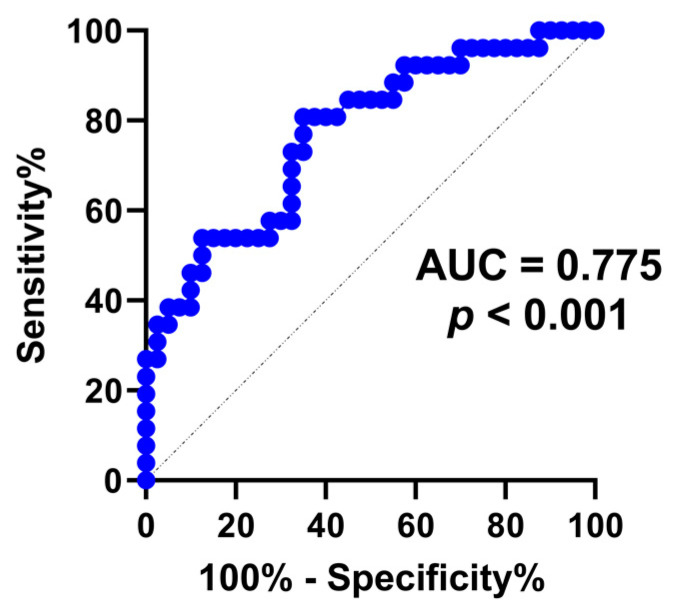
A ROC curve for predicting the improvement of renal function after surgery with urine NT-proBNP, AUC = 0.775 (0.65–0.89) and *p* < 0.001.

**Table 1 diagnostics-13-00247-t001:** Baseline characteristics.

	Grade of Hydronephrosis		
	Low Grade	High Grade	*p*-Value
No. pts	108	103	
Age (yrs)	52.8 (12.8)	56.2 (13.2)	0.06
Male sex (%)	60.2%	67.0%	
BMI (kg/m^2^)	25.9 (4.2)	28.2 (28.7)	0.41
Overweight (%)	56.5%	56.3%	
MAP (mmHg)	103.1 (14.1)	98.8 (14.5)	0.03
Hypertension (%)	60.2%	44.7%	
eGFR (mL/min/1.73 m^2^)	90.7 (23.3)	74.1 (28.7)	<0.001
CKD (%)	11.1%	33.0%	
Urine output (L/day)	1.2 (0.6)	1.3 (0.6)	0.46
Urine creatinine (mg/mL)	33.2 (31.9)	45.5 (26.8)	0.09
Stone size (cm)	1.1 (1.8)	1.0 (1.2)	0.46
Plasma NT-proBNP (ng/mL)	2.9 (1.8)	2.8 (2.8)	0.65
Urine NT-proBNP (pg/mg Cr)	11.1 (7.1)	6.7 (5.7)	0.006

The results are presented as mean (SD) or *n* (%) as appropriate. Abbreviation: pts, patients; yrs, years old; MAP, mean arterial pressure; eGFR, estimated glomerular filtration rate; CKD, chronic kidney disease; NT-proBNP, and N-terminal prohormone of brain-natriuretic peptide. Hypertension is defined as a mean awake systolic BP ≥ 130 mmHg and CKD is defined as an eGFR < 60 mL/min/1.73 m^2^.

**Table 2 diagnostics-13-00247-t002:** Comparison of the variables between the baseline and postoperative values. The statistical results were analyzed using a paired t-test and compared to the baseline in each group.

Variables	Baseline	Immediate	*p*-Value	Long-Term	*p*-Value
eGFR(mL/min/1.73 m^2^)	82.3 (27.1)	82.2 (26.7)	0.89	84.87 (27.79)	0.004
P NT-proBNP (ng/mL)	2.9 (2.3)	3.0 (2.3)	0.16	2.2 (3.1)	0.93
U NT-proBNP (pg/mL Cr)	9.4 (6.9)	15.0 (13.1)	<0.001	N/A	
Urine output (mL/day)	1244.8 (592.2)	1694.6 (771.7)	<0.001	1398.2 (338.5)	0.003

The results are presented as a mean (SD) or *n* (%) as appropriate. Abbreviations: eGFR, estimated glomerular filtration rate; P = plasma; U = urine; NT-proBNP, N-terminal prohormone of brain-natriuretic peptide; and N/A: not available.

**Table 3 diagnostics-13-00247-t003:** Comparison of variables between patients with improved or deteriorated renal function post-URS surgery.

		Immediate			Long-Term		
		eGFR ↓/ = (*n* = 136)	eGFR ↑ (*n* = 74)	*p*-Value	eGFR ↓/ = (*n* = 120)	eGFR ↑ (*n* = 74)	*p*-Value
Baseline eGFR		90.1 (24.9)	69.0 (26.4)	<0.001 *	89.4 (24.4)	72.4 (26.6)	<0.001 *
P NT-proBNP	Baseline	2.7 (1.8)	3.2 (3.2)	0.35	2.7 (1.7)	3.0 (3.2)	0.54
	△	0.2 (0.9)	−0.1 (1.0)	0.11	0.3 (0.9)	−0.03 (1.0)	0.11
U NT-proBNP	Baseline	10.3 (7.1)	7.2 (6.0)	0.08	11.1 (7.1)	6.7 (5.7)	0.006 *
	△	4.7 (14.5)	7.6 (15.5)	0.44	5.2 (16.8)	6.1 (11.0)	0.82

The results are presented as a mean (SD) or *n* (%) as appropriate. eGFR **↓/ =,** decreased or unchanged eGFR (deteriorated renal function), and eGFR ↑, increased eGFR (improved renal function). Abbreviation: P = plasma; U = urine; and △ = the difference of the variables between baseline and after surgery; * = statistically significant.

**Table 4 diagnostics-13-00247-t004:** A binary logistic regression model predicting odds of immediate renal function improvement after surgery.

	Univariate			Multivariate		
	OR	95% CI	*p*-Value	OR	95% CI	*p*-Value
**Sex**						
Female Male	1.0 (ref.)					
	1.4	0.65–2.13	0.30			
**BMI**						
Overweight Normal	1.0 (ref.)					
	0.87	0.48–1.57	0.65			
**Baseline CKD**						
Yes No	1.0 (ref.)			1.0 (ref.)		
	0.22	0.11–0.4	0.001	0.63	0.21–1.92	0.417
**Hydronephrosis**						
High Low	1.0 (ref.)					
	0.71	0.40–1.26	0.242			
**UTI**						
Yes	1.0 (ref.)					
No	1.44	0.81–2.58	0.215			
**Hypertension**						
Yes	1.0 (ref.)					
No	1.54	0.87–2.71	0.114			
**U NT-proBNP **						
High	1.0 (ref.)					
Low	3.34	1.19–9.38	0.022	3.24	1.09–9.70	0.035
**P NT-proBNP**						
High	1.0 (ref.)					
Low	0.92	0.73–1.14	0.43			

Abbreviations: OR = odds ratio; CI = confidence interval; BMI = body mass index; CKD = chronic kidney disease; UTI = urinary tract infection; P = plasma; U = urine; NT-proBNP = N-terminal prohormone of brain-natriuretic peptide. Urinary NT-proBNP was divided into high or low with a cutoff of 10 pg/mg Cr, and plasma NT-proBNP was divided into high or low with a cutoff of 0.2 ng/mL.

## Data Availability

Data are available on request due to restrictions. The data presented in this study are available on request from the corresponding author. Due to privacy concerns, the data is not publicly available.

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
