# Peer review of "Use of Urine N-Terminal Prohormone of Brain-Natriuretic Peptide (NT-proBNP) as a Non-Invasive Indicator for Renal Function Recovery after Surgical Relief of Hydronephrosis"

_diagnostics, 2023, doi:10.3390/diagnostics13020247_

Round 1

Reviewer 1 Report

Some methodological issues must be clarified. I suggest to request copy the approval ethical.

Author Response

Thank you for the suggestion. This is the copy of the approval ethical, the case-closing official document from the Institutional Review Board. The registration number “201205117RIC” is shown in this document.

Currently, we don’t have the English version of this document and it may take a while if you requested because we would have to file the application.

We've attached the case-closing official document from the Institutional Review Board, thank you. 

Reviewer 2 Report

The authors investigated the relationship between hydronephrosis and cardiorenal syndrome by examining the change in N-terminal prohormone of brain natriuretic peptide (NT -proBNP) in patients operated on to correct obstructive uropathy to predict recovery of renal function after correction of ureteral obstruction. They concluded that urine NT -proBNP could be a useful early marker of renal function recovery after URS surgery to identify patients whose renal and cardiac function were impaired by the obstruction.

I read the study with interest. Although the study is interesting, I noticed some major objections regarding the study design and methodology that, unfortunately, cannot be readily addressed by revisions. My concerns are as follows:

1. Title - It is not customary to include abbreviations in the title (NT -proBNP). Please include the full title and abbreviation in parentheses.

2. The authors stated that the data of this prospective study were collected between September 2012 and June 2013, but from the reference number we can infer that the study was approved by the local Ethics Review Board in 2021. Please provide the exact date of approval next to the registry number. How it is possible to collect samples prior to approval of Ethic Review Board? In addition, please explain why the authors waited 10 years from data collection to publication?

3. Since this is a prospective study, the authors should provide the sample size calculation in the methodology. I cannot see that the authors did a sample size calculation.

4. It is questionable whether the sample of patients in this study is representative-two important factors (duration of obstruction and comorbidities) may greatly affect the results and be a significant source of bias. Patients with comorbidities such as heart failure, anemia, or diabetes should also be excluded from the study.

5. Paragraph regarding laboratory investigation and how NT-proBNP was measured is not presented, it is unclear which kits and methods were used for analysis. How the samples were kept? It should be described in detail.

6. How the samples were analyzed, immediately or after collection of the whole cohort?

7. What statistical test was used to test normality of distribution of the data?

8. Table 1 - Please indicate the significance of the values in each row. It is unclear what each number represents. E. g. BMI – 25.9 (4.2) – What number in brackets represents? Measuring units for some variables are missing (e. g. BMI, MAP…). In addition, all abbreviations should be mentioned in a legend of each Table (e.g. BMI in Table 1…). The same follows for all Tables.

9. The manuscript should be proofread by an English editor or native speaker.

Author Response

Thank you very much for your valuable comment.

Reviewer 3 Report

The authors investigated the role of urine NT- proBNP as an indicator of renal function recovery after hydronephrosis surgery. The research idea is original and the results are interesting. Nevertheless, there are some major concerns regarding the presentation of the results:

1. The authors divided the patients in those with GFR improvement and GFR deterioration. Nevertheless, none of this notions were defined in the Material and Methods section. Please clarify.

2. The authors stated that lower urinary NT pro BNP was associated with renal function recovery but patients with low initial NTproBNP presented lower GFR at baseline. These results are conflicting. Given that baseline GFR was not the same in all patients it is difficult to interpret the effect of urinary NT proBNP in the renal recovery. The authors should divide the patients according to baseline GFR and perform subgroup analysis.

3. Moreover, it was noted that GFR change was negatively correlated to baseline urine NTproBNP. Again this result is difficult to interpretate, since it is possible that some patients presented normal GFR at baseline and that did not change but other presented lower GFR at baseline which was increased after the surgery. Please clarify.

4. In the logistic regression analysis in Table 4 the outcome was renal recovery or renal deterioration? If it is renal recovery and lower urinary NTproBNP was associated with better renal recovery the odds ratio should be less than 1 and not 3.24 as presented by the authors. Please clarify.

5. The authors stated that baseline GFR was not associated with NTproBNP. Please add the results of this association/correlation.

Author Response

Thank you for the valuable suggestions. Please see the attachment for our response, thank you very much!

Round 2

Reviewer 2 Report

The authors have improved the manuscript considerably. However, some of my objections have not been adequately addressed:

1. The authors should include a paragraph on sample size calculation. This is mandatory and I see no reason why they omit this in revised manuscript.

2. It is still unclear why the authors waited 10 years from data collection to publication? Please add a reasonable explanation.

Author Response

Thank you very much for your valuable comment. Please see the attachment.

Reviewer 3 Report

The authors have adequately responded to my comments. I have no further comments.

Author Response

(The authors gave the same response as above.)
